# Fairness by Learning Orthogonal Disentangled Representations

**Spyros Avlonitis**
University of Amsterdam
spyrosavl@gmail.com

**Alexander Papadatos**
University of Amsterdam
alexpapadatos95@gmail.com

**Danai Xezonaki**
University of Amsterdam
dxezonaki@gmail.com

## Reproducibility Summary

*In order to assess the reproducibility of the original paper, we implemented the models proposed and ran the experiments with the listed, in the paper, datasets. We implemented three encoder-discriminator models for the Tabular, CIFAR, and YaleB datasets. Due to missing information regarding the models' architecture, we had to incorporate some assumptions. In addition, for some of the models, we had to make assumptions regarding datasets' versions, preprocessing, and targets' definition. Our experiments for the German and Adult dataset approached the reported accuracies by the authors. However, we were not able to fully reproduce the results for CIFAR-10, CIFAR-100, and YaleB datasets.*

## 1 Introduction

In this paper we aimed to reproduce the results presented in Sarhan et al. (2020). In the original paper, the authors proposed a novel disentanglement approach to the invariant representation problem. They evaluated their approach on five publicly available datasets and compare it with state of the art methods for learning fairness and invariance.

## 2 Scope of reproducibility

In Sarhan et al. (2020), the authors proposed a novel model for learning a latent representation **z** that well describes a target observed variable **y** (e.g. Annual salary) while being robust against a sensitive attribute **s** (e.g. Gender or race). In order to achieve it, they decomposed the learned codes into sensitive and target representation. They imposed orthogonality and disentanglement constraints on the representations and force the target representation to be uninformative of the sensitive information by maximizing sensitive entropy.

The central claim the paper is that their approach performs better than state of the art models Sarhan et al. (2020) on three datasets (CIFAR-10, CIFAR-100, YaleB) and performed comparably on the other two (Adult, German).

In this work, we implemented the required models and reproduced the experiments as described in the original paper in order to validate their claims. Our methodology, models, datasets and results are described below in details.

## 3 Methodology

In this work, in the absence of an original implementation, we implemented the models based on the original paper. However, it should be noted, that some assumptions were made and are outlined in the following sections. The pseudocode of *Algorithm 1* in Sarhan et al. (2020) provided a low-level description of the key factors of each module and contributed in better understanding the proposed method. However, due to lack of information about the hyperparameters used in the original experiments, contacting the authors for more insights was vital for our work. For our experiments, we made use of GPUs of the Lisa cluster provided by surf.nl. Lastly, we documented all of our experiments with the help of Tensorboard.

### 3.1 Model descriptions

The architecture of all models implemented, consists of an encoding and a decoding part regardless the data used for each task. Our aim is to pass the input through an encoder which learns with the help of the decoder to produce two latent representations, $z_t$ and $z_s$. The objective of the encoder is to output a salient target representation $z_t$ while excluding sensitive information from it, which should be instead included in the sensitive representation, $z_s$. To this end, the encoding part consists of a shared encoder and two separate encoders for the target and sensitive representations, respectively. The input **x** is fed to the shared encoder, which distributes the produced output into the two separate

encoders. These in turn learn to produce the means and the logarithm of the standard deviations which parameterize the distributions of the target and sensitive representations.

Using the reparameterization trick, we sampled from the latent-space distributions and fed the sensitive representation to the sensitive discriminator and the target representation to both discriminators. Their objective is to minimize the divergence between the latent-space representations and the actual targets and sensitive attributes, respectively.

The required number of trainable parameters differs across datasets. Training our network on the German Credit and Adult Datasets required 21707 and 24715 trainable parameters respectively. For the CIFAR-10 dataset, our implementation required 11410828 parameters while for CIFAR-100 it required 11525112 parameters. Lastly, for Yale-B the network required 3300000 parameters.

## 3.2 Datasets

For each dataset, the authors have defined a set of sensitive attributes and a target task. In every case, the goal is to perform the target classification task while mitigating information leakage regarding the sensitive attributes.

**German Credit**   [Download link] This dataset classifies people described by a set of attributes as good or bad credit risks. It consists of 1000 instances and each one has 20 attributes. As the train/test split was not specified in the original paper, we randomly split the data to 900 samples for training and 100 for testing. the target task is to classify a bank account holder having good or bad credit risk. The sensitive attribute is the gender of the bank account holder.
There are not information regarding preprocessing in the original paper. In order to work with the dataset, we converted the categorical attributes into one-hot representations and normalised the numerals.

**Adult**   [Download link] The adult dataset contains 45,222 samples each with 14 attributes. The target task is a binary classification of annual income being more or less than \$50,000 and again gender is the sensitive attribute. 32561 samples in file *adult.data* were used for training, and 16282 samples in *adult.test* were used for testing.
There are not information regarding preprocessing in the original paper. In order to work with the dataset, we converted the categorical attributes into one-hot representations and normalised the numerals.

**CIFAR-10 & CIFAR-100**   [Download link] The CIFAR-10 dataset consists of 60000 32x32 colour images in 10 classes, with 6000 images per class. There are 50000 training images and 10000 test images. The CIFAR-100 dataset is just like the CIFAR-10, except it has 100 classes containing 600 images each. There are 500 training images and 100 testing images per class. The 100 classes in the CIFAR-100 are grouped into 20 superclasses. Each image comes with a "fine" label (the class to which it belongs) and a "coarse" label (the superclass to which it belongs).
In CIFAR-10 the 10 the target task is to classify the samples into *living* and *non-living*. The 10 fine classes are considered the sensitive attributes. In CIFAR-100 the coarse classes are used for the target task and the fine classes as sensitive attributes.

**Yale B extended**   [Download link] The extended Yale Face Database B contains 16128 images of 38 human subjects under 9 poses and 64 illumination conditions. The target task is the identification of the subject while the light source condition is considered the sensitive attribute.
The authors of the original paper do not specify which version of the dataset was used, original or cropped, and fails to provide any information regarding preprocessing. In addition, they do not specify how the sensitive classes are created. For our experiment we use the cropped version of the dataset and we split the various illuminations into 5 balanced classes based on which quadrant of the plane their azimuth and elavation fall into.

## 3.3 Hyperparameters

**Tabular datasets**   : Our model consists of three encoders, where the one is shared, and two discriminators. Following the setup of Sarhan et al. (2020), the encoders were implemented using one hidden layer of 64 units and the latent-space dimmensionality is set to be 2. The discriminators consist of two hidden layers with 64 units each. Models were trained for a maximum of 40 epochs and we used early stopping to select the model with the lowest validation loss. They were optimized using Adam with learning rate $10^{-3}$ and weight decay $5 \times 10^{-4}$.

**CIFAR dataset**   : The encoders were implemented using the ResNet-18 architecture He et al. (2016). The discriminators consisted of two hidden layers with 256 and 128 units, respectively. We used two separate optimizers for the encoding and the decoding part. Both networks were were optimized using Adam, with $10^{-4}$ learning rate and $10^{-2}$ weight decay for the encoders and with $10^{-2}$ learning rate and $10^{-3}$ for the discriminators. Once again, they were trained for a maximum of 40 epochs and we used early stopping to select the model with the lowest validation loss.

The hyperparameters $\lambda_{OD}$, $\lambda_E$, $\gamma_{OD}$ and $\gamma_E$ were set to the values specified in the supplementary material [1].

### 3.4 Experimental setup

Our experiments were implemented using PyTorch framework Paszke et al. (2019). For version controlling, we used a GitHub repository [https://git.io/Jt3R2]. Our models were trained using both CPU and GPU resources provided by the Dutch national e-infrastructure with the support of SURFCooperative.

### 3.5 Computational requirements

Due to the small amount of samples in the tabular datasets, namely the German Credit Dataset and the Adult Dataset, we were able to train our model on them using only a CPU. The former corpus required about 10 minutes to be trained on a CPU and 267 MB memory for training. The latter corpus required 15 minutes of training and 404 MB memory. For the CIFAR datasets, our experiments were also trained on a CPU. In particular, training our model on the CIFAR-10 and CIFAR-100 datasets took a total of 7 hours and memory of 556 and 430 MB, respectively. Lastly, in order to train the model on the Yale-B dataset a total of 171 MB memory were needed.

## 4 Results

In Table 1 we show the original accuracies reported in Sarhan et al. (2020) and our experiments' results.
Wee observe that were able to reproduce most of the reported results for the tabular datasets. In particular, we were able to almost accurately reproduce the results for the Adult dataset. For the German dataset, we were able to reproduce the target accuracy, whereas our implementation seemed to perform better on hiding sensitive information. Moreover, for the CIFAR-10, we were also able to approximate the target accuracy. However, we observe that our implementation allowed for more sensitive information leakage compared to the original results. Furthermore, we notice that for CIFAR-100 we could not reproduce or even approximate the reported accuracy values. Finally, although the model approximated the performance on the sensitive task for the Yale-B dataset, the corresponding target task could not be reproduced.

For our experiments we used the models as described in 3.1, the datasets in 3.2 and executed them with the hyperparameters listed in 3.3.

|  | Reported Accuracy | | Reproduced Accuracy | |
|---|---|---|---|---|
| Dataset | Target | Sensitive | Target | Sensitive |
| German | 0.7700 | 0.7100 | 0.7743 | 0.6519 |
| Adult | 0.8520 | 0.6826 | 0.8507 | 0.7166 |
| CIFAR-10 | 0.9725 | 0.1907 | 0.9445 | 0.4232 |
| CIFAR-100 | 0.7074 | 0.1447 | 0.1436 | 0.0720 |
| Yale B Extended | 0.8923 | 0.5292 | 0.6171 | 0.5782 |

Table 1: Original and reproduced experiments results.

## 5 Discussion

In this work, we reproduced the work presented in Sarhan et al. (2020). Since the code was not made publicly available, we implemented the proposed model from the beginning and thus we made assumptions for parts of the network and implementation. Our experiments showed that the reported results were approached for the German and Adult datasets. For the rest 3 datasets used (CIFAR-10, CIFAR-100, YaleB), we were not able to fully reproduce or even approximate the accuracy values, in some cases. Therefore, our experiments can support only the original paper's claim that concern the first 2 tabular datasets.

---

[1] `https://static-content.springer.com/esm/chp%3A10.1007%2F978-3-030-58526-6_44/MediaObjects/504495_1_En_44_MOESM1_ESM.pdf`

### 5.1 What was easy

Since the datasets used in the original paper are publicly available, an easy step of the reproduction process was obtaining the data. Moreover, the explanation of the two separate encoders and discriminators was sufficient in order to understand their architecture and put it into code. The overall concept and the motivation of the paper were made easy for the reader to understand in the Introduction section, without requiring knowledge on ensuring fairness in neural networks. Finally, communicating with the authors and getting feedback were easily achieved, as they responded quickly to our inquiries.

### 5.2 What was difficult

During the reproduction, one of the main difficulties that we faced was understanding the proposed network architecture. In particular, in the *Methodology* section of Sarhan et al. (2020), it is stated that the input **x** is fed to a shared encoder and then projected into two subspaces, which produce the learned target and sensitive representations. However, the architecture of the shared encoder is not defined in the provided implementation details. Instead, the authors describe only the architecture of the two separate encoders that produce the latent space representations. Furthermore, although it is stated that the Yale-B dataset consists of images for 38 individuals, the only available version online included 28 individuals. We experimented with both the original images and the provided cropped ones, and also removed some of the darker images in order to match the number of validation samples with the 1090 reported samples.

Another issue that was hard to understand was the training objective. Even though the authors provide in the pseudocode the mathematical formulas for some of the losses, the implementation of the KL-based losses and the overall training objective were confusing. Moreover, the authors do not provide information about the preprocessing method they perform, as well as the train-test split for the German Credit Dataset. As far as the training process is concerned, the original paper lacks in reporting the number of maximum epochs and whether the models were trained using cross-validation and early-stopping. In addition, the evaluation part of the models is unclear to the reader. Although in the *Experiments design* section, the authors claim that the target predictor has the same architecture as the discriminator, in the *Implementation details*, they provide different information for the target predictor used for tabular datasets.

### 5.3 Communication with original authors

After communcating with the authors, we received the paper's supplementary material [2], which was hard to find online. Moreover, they shared insights and information about back propagating the losses through the network, which made the training process easier to understand.

### 5.4 Acknowledgements

We would like to thank Stefan Schouten for giving us valuable insights into the task in hand throughout the reproduction process. Moreover, this work was carried out on the Dutch national e-infrastructure with the support of SURF Cooperative.

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
