# OpenReview forum: "Fairness by Learning Orthogonal Disentangled Representations"
_ML_Reproducibility_Challenge/2020 — Reject_

### Official Review · AnonReviewer1 · 2021-03-01
**Review of "Fairness by Learning Orthogonal Disentangled Representations"**

**Rating:** 5
**Confidence:** 4

**Review:**

Summary:
This work reproduces the main results of "Fairness by Learning Orthogonal Disentangled Representations". This includes implementing the model and evaluating it with different image datasets on downstream tasks.

Strengths:
* Evaluation of approach evaluated on 5 datasets

Weaknesses:
* I found the scope of this paper limited to the main results of the paper. At least, the authors should mention what is not included in the report. Further, it would have been good, if the authors had added an ablation study or the sensitivity analysis.
* It was not clear whether the hyperparameters used (3.3) were the same used in the paper or not.

**Familiar With The Original Paper:**

I have read the original paper

**Reproducibility Summary:**

Report has summary

---

### Official Review · AnonReviewer3 · 2021-03-01
**Fair information on reproduction of the original work**

**Rating:** 4
**Confidence:** 2

**Review:**

**Quality:** It is a fair quality report, which lacks on crucial details. Understanding of the authors on the original work was absent.

**Clarity:** The information presented is clear but the amount is less.

**Originality:** It is a normal report, nothing to fancy to be rated original.

**Significance:** The report currently is low on significance. It would have been better if the authors of the report provided some more details on how the original paper presented their works, and how the authors of the report see it and understand from their perspectives.

**Pros**

- Report covers all aspects of the reproducibility
- report of successful communication with authors of original work

**Cons**

- far less detailed to be called a report
- authors understanding and perspective of original work missing
- authors of the report just say that the results are dissimilar, however a much detailed discussion regarding the same giving reasons was felt missing
- overall, the report could have been improved with diagrams and figures added to enhance the information presented

**Familiar With The Original Paper:**

I have not read the original paper

**Reproducibility Summary:**

Report has summary

---

### Decision · Program_Chairs · 2021-03-31

**Decision:**

Reject

**Comment:**

Overall reviews and/or the paper content not good enough for the AC to recommend to the journal.